# The Relation between Korean Children’s Autonomy and Motor Development Mediated by Teacher–Child Relationships: A Focus on Gender Difference

**DOI:** 10.3390/ijerph192013527

**Published:** 2022-10-19

**Authors:** Yu-Jin Jang, Yea-Ji Hong

**Affiliations:** 1Department of Early Childhood Education, College of Social Science, Gachon University of Korea, Seongnam-si 13120, Korea; 2Department of Child Studies, Inha University, Incheon 22212, Korea

**Keywords:** autonomy, teacher–child relationship, fundamental movement skills, gender difference, whole-child development

## Abstract

This study was conducted to identify the relation between children’s autonomy and motor development mediated by teacher–child relationships. Are there differences between teacher–child relationships and motor development according to the gender of the child? To answer this question, the fundamental movement skills of 292 children were measured, and teacher–child relationship and children’s autonomy data were collected from the teachers. There was a gender difference in locomotion skills; however, there was no difference in object control skills. In the case of girls, a conflict teacher–child relationship mediates the association between autonomy and object control skills. This study highlights the importance of teacher–child relationships, which are mainly discussed in relation to conventional social-emotional development, and provides examples of whole-child development.

## 1. Introduction

The importance of good teachers cannot be overemphasized because early childhood teachers are the first people with whom children establish relationships outside of their families, and they are significant others who spend a large amount of time with children. Due to the increase in double-income families, children are entrusted to childcare centers at younger ages than ever before and stay in childcare centers longer. In Korea, more than 90% of children have attended kindergartens or childcare centers since 2012 [1]. The quality of teacher–child interactions is more strongly associated with child development than other measures of childcare quality [2]. Good teachers create an emotionally warm and supportive environment for children through mutual respect and positive communication and form positive teacher–child relationships by showing sensitivity to the childrens’ emotions.

The quality of teacher–child relationships is conceptualized through types of positive and negative influences [3]. Teacher–child relationships are formed through teacher–child interactions. Three broad domains, emotional support, classroom organization, and instructional support within teacher–child interactions are hypothesized to facilitate children’s developmental progress in the classroom [4]. Children who are emotionally supported perceive themselves and the environment surrounding them positively and act confidently. In a well-organized and well-structured classroom, children feel stable, use their energy appropriately, and participate appropriately in various activities. If a teacher supports children’s learning using an optimal method that suits the interests and developmental level of the children, they can acquire developmentally appropriate concepts, answer questions for themselves, and engage in effective learning. As a result, closer and higher quality teacher–child relationships are related to higher levels of academic, language and social-emotional development for the child [5,6,7,8,9,10]. On the other hand, young children who experienced more conflicts with their teachers had poorer language development and lower overall preacademic competence, higher levels of externalization and internalization problems, and more social adjustment problems [11,12].

The development of the whole child is a very important issue and the ultimate purpose of child education. It is known that boys’ perceptions of their physical exercise competence are better than those of girls, and that among all motor skills, their object control skills are also better than those of girls [13,14]. Children with good physical exercise skills have positive relationships with their peers, and the ability to perform well in sports and games is highly valued by children [15]. Since children play various games using their bodies, physical abilities are an important part of children’s play and peer relationships. Physically competent children have the opportunity to have fun with many friends, and it is possible that their social and emotional abilities may improve naturally. However, children with poor motor skills may begin to perceive themselves as less competent than their typically developing peers [16].

Fundamental movement skills (FMSs) are an indicator of children’s physical exercise ability and are an essential element of children’s physical play. FMSs are the foundation for more advanced movement sequences [17], are the most basic skills used in children’s daily lives and are evaluated qualitatively rather than quantitatively [18]. These skills are often described as ‘a set of processes associated with practice or experience leading to relatively permanent changes in the capability for movement’ [19]. FMSs comprise locomotion skills (i.e., moving the body from one location to another, e.g., running and jumping), object control skills (i.e., carrying or intercepting objects, e.g., throwing and kicking), and stability motor skills (i.e., having control over the body in both stationary and moving positions). In other words, FMSs are the basic component of more complex and advanced motor skills that enable children to continue participating in games, play, and daily life activities [20].

The difference of FMSs between boys and girls reports relatively consistent results; boys are more proficient than girls in object control from childhood [21,22]. This difference is more pronounced in adolescence, and only a few teenage girls are proficient in object control skills [21]. On the other hand, it is known that there is no difference between boys and girls in locomotion skills in childhood, and these results are also consistent with no gender difference in running [22], hopping [23], or jumping [22]. In previous studies, parental confidence in their skills to support their children to be active was only associated with object control skills and not locomotion skills. The authors predicted that children’s object control skills more rely on parental demonstrations and instructions than do locomotion skills [24]. Eventually, the significant others with children may affect the children’s physical development, and in this study, teachers who spend as much time with the children as parents will be considered.

The association between motor skills and psychosocial outcomes appears across the full continuum of motor ability [16]. According to a study that used machines to examine the relationship between the movement behaviors and development of toddlers for 24 h, the overall composition of movement was found to be important for children’s development [25]. Some evidence indicates that physical play, such as roughhousing, facilitates the neuroplasticity of certain brain structures and, as a result, improves cognitive function [26] and behavior [27]. Given that experiments with animals also revealed the relationship between physical play and the social brain [28], these results showed that there is a relationship between the movement behaviors of children and other aspects of development. Children with learning disabilities scored worse on both the locomotion and object control subtests than their typically developing peers [29]. Ultimately, children’s physical competence is expected to be deeply related to their cognitive and social-emotional competences. However, for teachers in the field of “early childhood education”, physical or exercise development has received less attention than cognitive and social-emotional development.

Although considerable research has been devoted to the association between teacher–child relationships and social-emotional competence or cognitive development, less attention has been given to the association with physical development. However, recent studies [16,30] examined the relationship between teacher–child relations and motor development. One result of the two-year survey of infants was the discovery that closeness and conflict in teacher–child relationships in the first year influence the motor skills of toddlers. The authors argued that improving the quality of the teacher–child relationships could promote the development of infants in this respect. With this, young children’s motor skills might be affected by environmental stress, such as the problems aroused from teacher–child and peer relationships [16]. The teacher–child relationship is an important variable that has a decisive influence on child development; however, its associations with motor development and with social-emotional development have not yet been clearly identified, so further research is needed.

The relation between teacher–child relationships and social-emotional competence has been discussed in many studies. Specifically, more positive teacher–child interactions in emotionally supportive environments and classrooms predicts more positive social interactions, a higher level of social competence, and fewer behavioral problems [6,9,31]. Autonomy, one of the social-emotional abilities of children, is discussed as being similar, in a sense, to initiative. Erikson (1975) argued in psychosocial development theory that if a child fails to acquire autonomy, she develops shame and doubt, and if the child fails to acquire initiative, she develops guilt [32]. According to Erickson (1975), between the ages of 3 and 6, children take the initiative to move through their play activities, expand their activity radius, and participate in various physical activities. Children’s autonomy is related to autonomy-supportive parenting and the teaching of significant others, such as parents and teachers [33,34,35,36]. Therefore, it can be assumed that autonomy is a variable that has an important influence on the development of children, including their motor development.

The experience of autonomy is expected to contribute to the child’s sense of being able to direct their own behaviors, to feel competent and accepted, and to develop committed compliance, accepting responsibility for compliance with values [37]. Children with a high level of autonomy can regularly engage and persist in various activities; thus, directly contributing to their self-regulation [38]. This connection is the same as for physical activity. A child’s motivation to participate in physical education is related to her FMSs; specifically, autonomous physical education motivation is related to balance skills [39]. According to a previous study, autonomy support from teachers has been positively and directly associated with students’ self-determined motivation for physical activity [40]. It is reported that autonomy improves the mental processes of the individual and increases satisfaction with activity, such as sports [41]. According to self-determination theory, autonomy is a significant factor that underpins psychological health and effective engagement with the world [42]. Ultimately, autonomy is expected to be one of the variables that influences the motor development of children.

The teacher–child relationship is closely related to support for the child’s autonomy [7,43]. Close teacher–child relationships may provide ample opportunities for children to explore their surroundings, practice their self-help skills, and gain confidence in their ability to try new things with autonomy [44]. Autonomous children who control themselves have a good relationship with teachers, and good teacher–child relationships have a positive effect on children’s development. This relationship can be explained by the bidirectional model. Within a secure relationship, teachers could provide responsive care during daily routines, and toddlers are able to better develop motor skills in indoor and outdoor play [30]. Meanwhile, the teacher–child relationship differs depending on gender. Toddler girls, compared to boys, were more likely to seek contact from their caregivers and stay in close proximity to regulate distress [45]. In contrast, boys were more likely to experience conflicts with their teachers than girls [46]. Therefore, it is necessary to consider the gender of children in discussions related to teacher–child relationships and autonomy.

Korea is a country well known for its high level of education, which has various effects on the development of children. Many Korean children have been exposed to the academic environment since childhood; therefore, there is not much play time and free time for children. According to the research report of the KICCE (Korea Institute of Child Care and Education), 83.6% of five-year-old children were found to receive private education. It was also found that five-year-old children are taking 2.2 kinds of private education [47]. Children who are required to study in their daily lives and who cannot control their daily time are expected to have lower autonomy. A sociocultural atmosphere that does not allow for autonomy in such early stages may result in negative consequences for these children. Actually, Korean children’s need for autonomy were considerably lower than other psychological needs [35]. This study predicted that the autonomy of children would be related to the development of FMSs, and that the association would be mediated by the teacher–child relationship.

This study aims to identify the relation between Korean children’s autonomy and physical development, especially motor development mediated by teacher–child relationships. Based on previous studies, it was considered that there is a gender difference between children’s motor development and teacher–child relationships. This study was conducted to confirm that social-emotional development was also related to the physical development of children, and that the teacher–child relationship was also related to the physical development of children.

## 2. Methods

### 2.1. Participants and Data Collection

The sample included 292 children (151 boys and 141 girls) and 14 teachers who were recruited from four childcare centers in Seoul, Korea. All of the teachers in this study had an educational background above that of a junior college, which is the same as that of a general kindergarten teacher. The investigation was conducted 9 months after the children met their new homeroom and physical education teachers. If the mother agreed to participation in the study, we sent a questionnaire and measured the child’s physical development with the test of gross motor development second edition (TGMD-II) at the childcare center the child attended. In this study, the children were 3 (N = 95), 4 (N = 100), and 5 (N = 97) years old A total of 71.6% of the mothers were in their 30s, and 44.8% had a university degree. Using the questionnaire, the teachers provided data on the children’s social competence and the teacher–child relationship. Physical development was measured directly by the physical education teachers at the children’s childcare centers. A single investigator with 10 years of experience as a kindergarten physical education teacher and a master’s degree in motor development and learning scored the TGMD-II results.

### 2.2. Measures

#### 2.2.1. Teacher–Child Relationship

In this study, teacher–child relationships were measured by the Student–Teacher Relationship Scale (STRS) [48]. The STRS assesses teachers’ perceptions of their relationships with students via teacher reports. It has two subscales, closeness and conflict. The closeness subscale assesses teachers’ perceptions of the warmth and openness of the communication with the child (e.g., “This child openly shares his or her feelings and experiences with me”). The conflict subscale assesses the degree of hostility and non-cooperation in interactions (e.g., “this child easily becomes angry with me”). We used the short form of the STRS with 15 items (8 items for closeness and 7 items for conflict). Teachers rated each item on a 5-point scale (1 = “definitely does not apply” to 5 = “definitely applies”). The reliability and validity of this measure have been established by various studies [7,38,47]. The Cronbach’s α was 0.82 for closeness and 0.83 for conflict.

#### 2.2.2. Autonomy

To measure the children’s autonomy, we used the Devereux Early Childhood Assessment (Devereux Early Childhood Assessment: DECA) [49]. In many prior studies from various countries [49,50,51], the DECA was used to measure the social-emotional competence of 2- to 5-year-olds. The DECA consists of a total of 37 items, with 27 items making up the Total Protective Factors (TPF) scale and 10 items making up the Behavioral Concerns (BC) scale. The subfactors for the TPF scale are initiative, self-control, and attachment, which are typical social-emotional competence variables. In this study, we used only the initiative questions.

The initiative subscale measures a child’s ability to think independently, make choices, and carry out actions to meet his or her needs [52]. The initiative subscale includes not only independent processing and decision making but also persisting even if what is being done does not work out, trying new activities, and focusing on tasks or activities. In other words, it includes positive motivation to continue activities. Because of the subtle differences in the meaning of the terms, we identified initiative with autonomy [53,54,55]. A five-point Likert scale was used; the higher the score was, the higher was the level of autonomy. The Cronbach’s α was 0.90 for autonomy.

#### 2.2.3. Fundamental Movement Skills

To measure children’s FMSs, the TGMD-II [56] was used, as it was specifically designed and validated for use with children aged 3–10 years. It is a process-oriented assessment that incorporates the qualitative aspects of movement behaviors [14]. The TGMD-II is a valid and reliable evaluation tool that measures motor ability in children and has been used in many studies [14,57,58]. In this study, the FMSs of children were recorded using two cameras, and the FMS scores were assessed via video analysis.

This test has two subsets for a total of 12 items: locomotion (i.e., running, galloping, sliding, hopping, leaping, and horizontal jumping) and object control skills (i.e., overhand throwing, underhand rolling, striking a stationary ball, stationary dribbling, kicking, and catching). The six items for locomotion skills include 26 subitems, and the six items for object control skills include 24 subitems. The children received 1 point if a criterion was met and 0 if not. After the children performed each of the six subskills for locomotion and object control skills twice, a single investigator gave a score of l or 0 for each criterion. Therefore, the total score for the locomotion and object control skills ranged from 0 to 48 points. Since the number of tasks for each set of skills is six, the final score ranges from zero to eight after dividing the total score by six.

### 2.3. Analyses

The IBM SPSS Statistics version 22 was operated to analyze descriptive statistics and intercorrelations. A t-test was conducted to determine the difference in FMSs according to the gender and age of children. A Pearson correlation analysis was used to examine the associations between the main variables. To examine gender differences, we conducted a multi-group analysis to identify whether the path coefficients significantly differed between boys and girls. An unconstrained model and a constrained model were used to compare the gender difference with AMOS 24.0 program.

## 3. Results

### 3.1. Preliminary Analyses

The *t*-test was performed according to age and gender to clearly identify the age at which the difference in FMSs by gender of the children appears. FMS scores by age and gender are shown in Table 1. Locomotion skills and object control skills scores increased with age. The object control skills score of all ages was lower than the locomotion skills score. After examining the differences by gender, the locomotion skills of 5-year-old boys and girls were found to be similar, so the differences by gender were not significant (*t* = 1.70, n.s.). However, in the case of object control skills, the boys’ scores were significantly higher than the girls’ scores, and the difference was particularly noticeable at the age of 5 (*t* = 6.29, *p* < 0.001). According to our data, the gender difference in object control skills in children is not statistically significant, but the gender difference becomes statistically significant at age 4.

The means and correlation coefficients for the children’s social-emotional competence, teacher–child relationships, and FMSs can be found in Table 2. Teacher perceptions of the children’s autonomy was 3.31 points (range: 0–5), and that of the children’s self-regulation was 3.04 points (range: 0–5). Perceptions of closeness in the teacher–child relationships were higher than the median (M = 3.30), and those for conflict were much lower than the median (M = 3.30). Korean children’s average locomotion skills score was 4.96 points (range: 0–8), which is higher than the median; however, their object control skills score was 3.61 points (range: 0–8), which is lower than the median.

There was a correlation between autonomy and both types of FMSs: locomotion skills (r = 0.20, *p* < 0.01) and object control skills (r = 0.18, *p* < 0.01). However, self-regulation was related to locomotion skills (r = 0.11, *p* < 0.05), but not to object control skills (r = 0.05, n.s.).

The level of conflict in teacher–child relationships and the FMSs (locomotion skills (r = −0.15, *p* < 0.01) and object control skills (r = 0.10, *p* < 0.05)) were negatively correlated. The design phase of the study predicted that the level of both closeness and conflict in teacher–child relationships would be related to FMSs. However, there was no correlation between the closeness of the relationships and locomotion skills or object control skills.

### 3.2. Gender Differences

The results of the *t*-test showed that there were significant gender differences in the object control skills, and boys scored higher than girls. To further examine gender differences, multi-group analysis was conducted to identify whether the path coefficients significantly differed. We used two models to compare the gender difference according to Byrne (2001): (1) an unconstrained model. Allowing all the paths to vary across male and female groups; (2) a constrained model, constraining all the parameters to be equal across boy and girl groups. The results showed that differences between these two models were not significant (Δχ^2^ = 2.460, df = 3, RMSEA = 0.054, NFI = 0.952) in locomotion skills, and partially significant (Δχ^2^ = 5.251, df = 3, RMSEA = 0.057, NFI = 0.905) in object control skills, based on the critical ratios of differences (CRD) of recommendation of Arbuckle (2003).

While applying equivalent constraints to each path, it was confirmed whether there was a difference in the model fit and χ^2^ value of the model in Table 3. Among the constrained model of object control skills, the entire constrained path model is rejected; however, it can also be confirmed that there is a path in which partially equivalent constraints are established. The model of locomotion skills, based on the difference in χ^2^ values according to the degree of freedom, all of the paths (autonomy→T-C conflict, T-C conflict→LC, autonomy→LC) were not significantly different. On the other hand, in the model of object control skills, two paths (T-C conflict→LC, autonomy→LC) were significantly different. The path coefficient of child autonomy to object control skills for boys (β = 0.34, *p* < 0.001) was significant; however, not for girls (β = 0.08, n.s.). The path coefficient of teacher–child conflict to object control skills for girls (β = −0.21, *p* < 0.05) was significant; however, not for boys (β = −0.04, n.s.). The results reported that object control skills of boys show a different aspect from FMSs of girls and locomotion skills of boys.

As a result, in both boys and girls, children’s autonomy influenced locomotion skills through the teacher–child conflict relationship. In the locomotion skills, the difference of path according to gender was not significant (Figure 1). On the other hand, in the case of object control skills, the autonomy of boys influenced the teacher–child conflict relationship and object control skills, but the mediating effect of the teacher–child conflict relationship was not significant in the effect of boy autonomy on object control skills (Figure 2). In the case of girls, the same pattern as the locomotion skills appeared, and the girl’s autonomy influenced the locomotion skills through the teacher–child conflict relationship (Figure 3). In the locomotion skills of boys and girls and the object control skills of girls, the teacher–child conflict relationship was found to completely mediate the effect of a child’s autonomy on FMS skills.

## 4. Discussion

This study was conducted to identify the relation between Korean children’s autonomy and motor development mediated by teacher–child relationships. We considered that there is a gender difference among children’s autonomy, motor development, and teacher–child relationships. To this end, data on children’s autonomy and teacher–child relationships were collected through questionnaires. In addition, a professional physical education teacher measured the children’s FMSs. The main results and discussion follow.

First, children’s locomotion skills and object control skills increased with age, and the gender differences varied according to the FMS subfactors. There were no gender differences in the children’s locomotion skills, and boys had higher object control skill scores at all ages. These results are partially consistent with those of previous studies [13,14]. The higher object control skill scores for boys than for girls has also been consistently confirmed in many previous studies of children and adolescents [13,14,59]. Researchers have argued that these results are due to environmental factors. Before adolescence, in early childhood, girls do not have worse physical abilities than boys. In contrast with girls, boys have environmental and emotional motivations to acquire object control skills. Boys participate in more offensive play with balls and focus on achieving goals, while girls display more off-task or spectator-player behaviors [60]. For this reason, the locomotion skills of girls are not different from that of boys, while the object control skills are different from childhood. One possible explanation for this gender difference is probably because support for sports equipment varies by gender, and this support helps develop physical competence [24]. Boys generally play more ball games or physical games using equipment than girls from childhood, and parents provide more equipment for children with better locomotion and object control skills [61]. A boy with more sports equipment will play more physical activities, and this cycle will likely repeat and interact over a long time and increase his competence to manipulate objects.

Second, as shown in the descriptive statistics, there is a big difference in FMSs depending on a child’s age, and the locomotion skills score was higher than that for object control skills [13,14,39]. Object control skills are more difficult than locomotion skills. Object control skills are advanced skills for children that are expected to require more specific muscle development than locomotion skills. In childhood, gross motor skills develop quickly; however, fine motor skills are still underdeveloped. In order to control objects, such as balls, fine motor skills must be used together with gross motor skills, and more sophisticated movements are required. The fact that the locomotion skill score was higher than the object control skill score reflects this development.

Third, according to a result of multi-group analysis, there was no gender difference in locomotion skills. However, in terms of object control skills, the gender difference between boys and girls was significant. Children’s (boys and girls) autonomy influenced locomotion skills through the teacher–child conflict relationship, and girls’ autonomy influenced object control skills through the teacher–child conflict relationship. On the other hand, boys’ autonomy influenced the teacher–child conflict relationship and object control skills, but the mediating effect of the teacher–child conflict relationship was not significant on object control skills. These results imply that the association of children’s social-emotional development and teacher–child relationship to motor development, suggest the possibility that conflict relations with teachers mediate various aspects of child development. These results support the results of previous studies [16,30] that the conflicting teacher–child relationship is negatively related to the physical development of a child. Specifically, even if the child’s autonomy is somewhat low, if the relationship with the teacher is not negative, there is room for the development of the boys’ and girls’ locomotion skills and the girls’ object control skills. On the other hand, it was found that boys’ object control skills were not related to the teacher–child relationship, and in the case of boys, it suggests that personal motivation or environmental factors may affect the development of object control skills rather than the teacher–child relationship. It is worth remembering that motor skill in pre-school aged children is multidimensional, being associated with factors at the child, family and environmental level [24].

Based on these results, we drew the following conclusions. First, this study provides empirical evidence for whole-child development. Physical education classes are organized as independent classes in kindergartens and child care centers in Korea, and most of these classes are conducted by professional physical education teachers. Therefore, there is a slight possibility that homeroom teachers are not skilled in the development of motor skills in children. In this study, physical education teachers measured the motor development, and the social-emotional development was measured by the homeroom teachers. Nevertheless, the fact that children’s motor development is related to social-emotional competence and to the quality of the teacher–child relationships as reported by homeroom teachers is evidence of whole-child development. Early childhood education teachers need to be as interested in the physical development of children as in their social-emotional and cognitive development.

Second, among the FMSs, object control skills were found to be more difficult to achieve than locomotion skills, and more difficult for girls. Object control skills require careful movements of the body. Of the two motor skills, locomotion skills tend to develop naturally with age, unlike object control skills. In other words, it is predicted that more activities or training are needed for children to develop object control skills. In particular, girls need various activities that can help them to develop their object control skills and motivate them to acquire those skills. It is necessary to plan various activities for girls to play with various objects, such as balls, and to devise team games for girls in kindergartens and schools. Mixed games with boys and girls would be okay with locomotion skills; however, in object control skills, playing with boys is likely to dampen girls’ motor activities and motivations. Therefore, if girls’ object control skills are low, it is recommended to separate gender in the early stages of physical activity.

Third, the level of conflict in teacher–child relationships is not only negatively related to children’s autonomy but also negatively related to children’s motor development. Since this study is not longitudinal, it is difficult to predict the effect of autonomy and teacher–child relationships on motor development; however, this should be discussed in bidirectional models of teacher–child relationships [62]. Bidirectional models are based on transactional theory [63], which assumes that children’s developmental outcomes are the product of a combination of an individual child’s characteristics and experiences and aspects of his or her environment (including teacher–child relationships). Children who move and control their bodies at an appropriate level also engage in various activities in the classroom and are not in conflict with their teachers. In addition, nonconflictual relationships with teachers give children a sense of stability and allow them to accept themselves and the environment around them. In addition, such children participate properly in various activities and properly use their energy in appropriate relationships with their teachers.

A low-conflict relationship between teachers and children affects the overall development of children, and the overall development of children also affects the teacher–child relationship [5,7,8,10]. Ultimately, the physical development of children needs to be discussed as much as their social-emotional and cognitive development in terms of their relations with the teacher–child relationship, and more attention and effort needs to be directed towards the physical development of children. Regarding physical development, positive teacher–child relationships were not significantly correlated, while the negative aspect of those relationships was found to be highly correlated. This finding requires further research. Teachers should create a non-conflict teacher–child relationship for the whole development of children. Teacher–child relationship, which are not conflicting for both boys and girls, benefit children not only in their social-emotional development but also in terms of physical development. In addition, it is necessary to provide various physical activities, such as balls and objects to use in outdoor play spaces, during the daily life of kindergartens or daycare centers. In addition, modeling female teachers’ physical activity will have a positive effect on the FMSs of children, especially girls. The demonstration of physical activity behavior by significant others (i.e., parents and physical education teachers) is expected to attract individuals within the same environment to initiate physical activity behaviors [64].

In this study, among the social-emotional development scale (DECA), only child autonomy was related to a child’s FMSs, and the relationship with self-regulation was not significant. As a result, we excluded variable of self-regulation from the analysis. We believe that these results are related to the attributes of the scale and the polymorphism inherent in the concept of self-regulation. In this study, items on the self-regulation scale consisted of questions related to behaviors, such as refraining from expressing emotions and cooperating in relationships with others or in social contexts. Self-regulation is largely divided into its emotional, behavioral, and cognitive aspects, and on this scale, self-regulation might be closest to its emotional dimension. If self-regulation were considered in terms of behavior, such as impulse control or delayed gratification behaviors, it is predicted that the association with FMSs would be significant.

As the introduction revealed, children in large Korean cities (i.e., Seoul) are likely to have lower autonomy compared to children in other countries. Therefore, it is necessary to examine the association between autonomy and physical development of children in other countries. The development of autonomy and FMSs is an important issue for all children around the world. This study provides evidence that children’s autonomy levels and teacher–child relationships are related to FMSs. However, there is a possibility that the autonomy of children may vary depending on the cultural situations, and the teacher–child relationship may also differ between the West and the East countries. In general, the East tends to promote more relatedness rather than autonomy, and the West is likely to have individualistic values that promote more independence [65,66]. Therefore, in cases of Western children with high autonomy, it is necessary to check whether the relationship between teacher–child relationship and FMSs is different from that of Korean children. 

## 5. Conclusions

This study provides empirical evidence for whole-child development. The object control skills were found to be more difficult to achieve than locomotion skills, and more difficult for girls. The level of conflict in teacher–child relationships is not only negatively related to children’s autonomy but also negatively related to children’s motor development. Since the social-emotional development of children has various aspects and can be discussed as various variables, further research is needed on the association between social-emotional development and FMSs. It is a limitation of this study that the teacher–child relationship was measured by a teacher’s questionnaire. In future studies, it is necessary to measure the teacher–child relationship by observing kindergarten classrooms and examining the association with children’s FMSs. This study suggests that the importance of the teacher-child relationship that mediates the association between child autonomy and physical development and the gender of the child should be considered.

## Figures and Tables

**Figure 1 ijerph-19-13527-f001:**
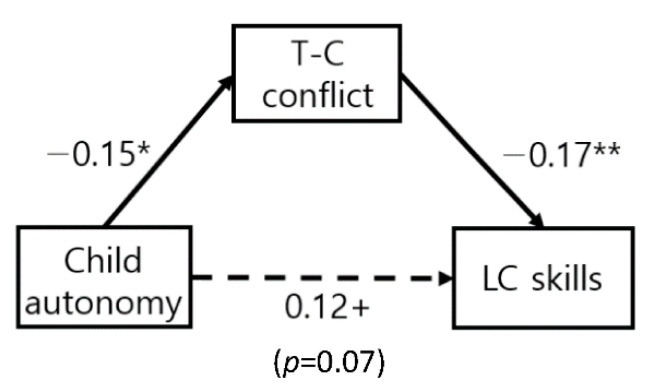
Locomotion skills of boys and girls (no gender difference). ^+^
*p* < 0.10, * *p* < 0.05, ** *p* < 0.01.

**Figure 2 ijerph-19-13527-f002:**
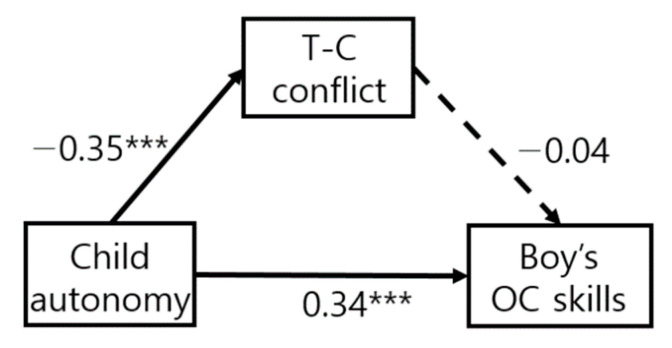
Object control skills of boys. *** *p* < 0.001.

**Figure 3 ijerph-19-13527-f003:**
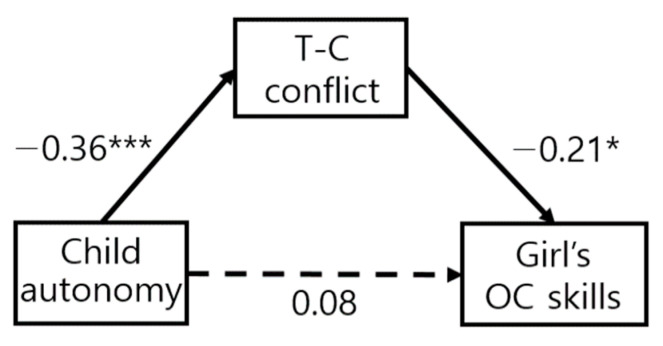
Object control skills of girls. * *p* < 0.05, *** *p* < 0.001.

**Table 1 ijerph-19-13527-t001:** FMS score by age and gender (n = 292).

		M (SD)			M (SD)	*t*
Fundamental Movement Skills	Locomotion skills	3 years (n = 95)	4.27 (1.12)	boy	4.78 (1.27)	0.42
				girl	4.66 (1.51)	
		4 years (n = 100)	5.18 (1.33)	boy	5.14 (1.17)	−0.27
				girl	5.21 (1.51)	
		5 years (n = 97)	6.22 (0.66)	boy	6.32 (0.67)	1.70
				girl	6.08 (0.65)	
	Object control skills	3 years (n = 95)	2.71 (1.06)	boy	3.19 (1.03)	2.13
				girl	2.76 (0.91)	
		4 years (n = 100)	3.43 (1.15)	boy	3.70 (1.27)	2.48 *
				girl	3.13 (0.95)	
		5 years (n = 97)	5.28 (1.35)	boy	5.87 (1.33)	6.29 ***
				girl	4.45 (0.87)	

* *p* < 0.05, *** *p* < 0.001. Notes. M, mean; SD, standard deviation.

**Table 2 ijerph-19-13527-t002:** Correlation of main variables.

	1. Autonomy	Teacher–Child Relationship		FMSs	
		2. Closeness	3. Conflict	4. L-M Skills	5. O-C Skills
1	1				
2	0.62 ***	1			
3	−0.35 ***	−0.33 ***	1		
4	0.20 **	0.04	−0.15 **	1	
5	0.18 **	−0.08	−0.10 *	0.66 ***	1
M (SD)	3.31 (0.79)	3.30 (0.82)	1.82 (0.78)	4.96 (0.80)	3.61 (1.24)

* *p* < 0.05, ** *p* < 0.01, *** *p* < 0.001. Notes. M, mean; SD, standard deviation; L-M, locomotion; O-C, object control.

**Table 3 ijerph-19-13527-t003:** Comparison of differences with base model.

	The Path of Equivalent Constraints	Δdf	Δχ^2^	ΔNFI	ΔCFI
LC skills	autonomy → T-C conflict	1	0.001	0.000	0.012
	T-C conflict → LC	1	1.180	0.023	0.00
	autonomy → LC	1	2.091	0.040	0.002
OC skills	autonomy → T-C conflict	1	0.001	0.00	0.023
	T-C conflict → OC	1	5.138 *	0.093	0.073
	autonomy → OC	1	3.985 *	0.072	0.520

* *p* < 0.05. Notes. df, degree of freedom; NFI, normed fit index; CFI, comparative fit index.

## Data Availability

This data was collected privately, therefore it was not disclosed.

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
