# Peer review of "The Relation between Korean Children’s Autonomy and Motor Development Mediated by Teacher–Child Relationships: A Focus on Gender Difference"

_ijerph, 2022, doi:10.3390/ijerph192013527_

Round 1
Reviewer 1 Report
Firstly, I would like to congratulate the authors on this outstanding study. It is so necessary for the literature about child development.
Generally, I consider the manuscript excellent and worth it for publication in the Journal. However, before I recommend the paper for publication I have some minor concerns to be clarified and amends to improve the manuscript.
The introduction is well written but too long. The authors have to check repetitive ideas over the paragraphs and prioritize the ones that are adding strong rationale to the research question. Otherwise, it will be extremely exhausting for readers.
Table 1 – Please include the sample size for each child age subgroup;
Table 2 – What are the numbers (1-5) in the left column? Please explain in the table.
Why the authors are analyzing motor performance through TGMD according to child’s age subgroups? What is the relevance of that, since the test already classifies the scores based on age? Please, explain!
Discussion (Page 9) – Please explain in more detail what are the environmental factors you have mentioned that are justifying the differences between boys and girls;
Finally, I did not see the limitation section at the end of the discussion. Please, include!
Author Response
Dear Editor
We are enclosing a revised manuscript entitled, “The relation between Korean children’s autonomy and motor development mediated by teacher-child relationships: Focus on gender difference”. The manuscript is 14 pages long and includes 3 tables and 3 figures.
We appreciate your thorough review of our manuscript. The comments were helpful to us in revising the paper. Our revised manuscript addresses and incorporates each of the concerns and suggestions raised by the reviewer. We believe that your recommendations have greatly improved the presentation and organization of our research study. We thank you for bringing up all the points.
We revised the following contents.
- According to the first reviewer’s suggestion, we deleted approximately half of the volume of introduction. By deleting the contents about infants and adolescents’ suggested by the second reviewer, we were able to mainly focus on discussing those of children in the current study.
- According to the first reviewer’s suggestion, we added not only the reason for analyzing exercise performance through TGMD according to the age group, but also the limitation of the current study. In addition, the explanation of environmental factors that justify the difference between boys and girls was discussed further.
- According to the second reviewer’s suggestion, the information about teachers participated in the current study was added and some ambiguous sentences were deleted.
- According to the second reviewer’s suggestion, the analysis section and paragraphs on international implications were added.
Thank you for your consideration of this manuscript. Please address all correspondence concerning this manuscript to me at yjhong@inha.ac.kr

Reviewer 2 Report
Abstract
1. The abstract of the study is well - structured and well - written. It is one of the most important presupposition in order to attract reader to continue reading the manuscript.
Introduction
2. kindergartens and childcare centres are not the same. Are there elements for each case?
3. Additionally, we need to know who is the teacher at those centres in the case of Korea. Otherwise if you are talking about teachers in general and the respective teacher - child relationships you have to use a different paragraph.
4. Page 2”…move and play, especially for boy”. Why especially for boy? I believe that authors have to declare the age they are talking about.
5. it is difficult for me to understand if in the case of infants aged 1 we are talking about teachers. If this is common in Korea authors have to explain further the respective qualifications.
6. In general, the literature review is well structured, with important recent references.
Methodology
7. At the methodology the age of the children is clear. I believe that the theoretical framework has to be concentrated only on those ages, otherwise many questions are raised.
8. The presentation of the methodology is clear. However I believe that the limitations which derived by the methodology could be presented at this section.
9. It would be helpful to have a section about the analyses, as well.
Results
10. The presentation of results is excellent
Discussion
11. The discussion is clear in respect to the educational system of Korea. A paragraph could be added for the international implications.
Author Response

(The authors gave the same response as above.)
